# Motor engagement relates to accurate perception of phonemes and audiovisual words, but not auditory words

Kelly Michaelis [1,2], Makoto Miyakoshi[3], Gina Norato[4], Andrei V. Medvedev[1] & Peter E. Turkeltaub[1,5 ✉]

A longstanding debate has surrounded the role of the motor system in speech perception, but progress in this area has been limited by tasks that only examine isolated syllables and conflate decision-making with perception. Using an adaptive task that temporally isolates perception from decision-making, we examined an EEG signature of motor activity (sensorimotor μ/beta suppression) during the perception of auditory phonemes, auditory words, audiovisual words, and environmental sounds while holding difficulty constant at two levels (Easy/Hard). Results revealed left-lateralized sensorimotor μ/beta suppression that was related to perception of speech but not environmental sounds. Audiovisual word and phoneme stimuli showed enhanced left sensorimotor μ/beta suppression for correct relative to incorrect trials, while auditory word stimuli showed enhanced suppression for incorrect trials. Our results demonstrate that motor involvement in perception is left-lateralized, is specific to speech stimuli, and it not simply the result of domain-general processes. These results provide evidence for an interactive network for speech perception in which dorsal stream motor areas are dynamically engaged during the perception of speech depending on the characteristics of the speech signal. Crucially, this motor engagement has different effects on the perceptual outcome depending on the lexicality and modality of the speech stimulus.

[1] Center for Brain Plasticity and Recovery, Georgetown University Medical Center, Washington, DC, USA. [2] Human Cortical Physiology and Stroke Neurorehabilitation Section, National Institute for Neurological Disorders and Stroke (NINDS), National Institutes of Health, Bethesda, MD, USA. [3] Swartz Center for Computational Neuroscience, Institute for Neural Computation, University of California San Diego, San Diego, CA, USA. [4] Clinical Trials Unit, National Institute of Neurological Disorders and Stroke, National Institutes of Health, Bethesda, MD, USA. [5] Research Division, Medstar National Rehabilitation Hospital, Washington, DC, USA. ✉email: turkeltp@georgetown.edu

While speech perception primarily occurs via temporal lobe pathways[1–4], there is now a wealth of evidence that frontal lobe motor areas typically responsible for speech production are also active during speech perception[5–8] (see ref. [9] for a review). However, the role of these motor areas in speech perception remains a matter of controversy, with some researchers concluding that motor regions play at most a minor supporting role[2,10,11], and others suggesting that these areas are essential for perception[5,12,13]. There are several possible explanations for why motor areas activate during speech perception, and understanding the specific stimulus and task contexts in which these areas are engaged is a crucial step in evaluating these competing hypotheses. In the context of the Rauschecker and Scott dual-stream model[1,4,14,15], the present study proposes specific hypotheses regarding the conditions under which motor areas are flexibly engaged for speech perception. Using a novel behavioral task combined with electroencephalography (EEG), we test these hypotheses while addressing critical gaps in the literature.

The dominant explanations for motor involvement in speech perception stress the importance of articulatory motor plans, suggesting that the motor system helps solve the "lack of invariance" problem in speech perception by using representations of motor plans to constrain the interpretation of incoming information[9,12,16–19]. The idea that representations of speech sounds in motor areas (including primary motor cortex, premotor cortex, and Broca's area (BA44/45)) provide a template for decoding incoming input is especially common in studies investigating noisy or degraded speech[12,16,17,20–23]. These motor modeling theories lead to two predictions: (1) that motor activity during perception should be specific to speech or other "doable sounds" that we have practiced articulatory plans for producing, and (2) that motor involvement depends on the task context and stimulus features. In contrast to motor modeling explanations, alternative hypotheses suggest motor activity is not specific to speech perception but instead reflects more domain-general processes like attention, decision making, or early covert rehearsal[10,24,25].

In order to adjudicate between these competing hypotheses and address critical gaps in the literature, we designed a behavioral task that investigates four key stimulus features potentially influencing motor activity during speech perception: (1) whether the stimulus is speech or a non-doable, non-speech sound, (2) whether the speech is lexical or sublexical, (3) whether the speech is auditory-only or audiovisual, and (4) whether the signal is clear or challenging to perceive. Importantly, this task also allowed us to investigate whether motor processing is specific to speech perception, or if, as alternate hypotheses suggest, it reflects more domain-general processes[10,24,25]. Examining these stimulus features addresses several unanswered questions. First, as mentioned above, the dominant explanations for motor activity during perception rely on modeling of internal speech representations, suggesting that motor activity should be specific to sounds that we have articulatory plans for producing. However, few studies have directly addressed the question of speech specificity, and the existing results have been inconsistent[26,27]. Second, prior studies have almost exclusively examined phoneme or syllable perception as opposed to whole words, with the few exceptions producing ambiguous results[28,29]. In order to understand whether the motor system participates in perception in real-world scenarios, it is crucial to examine motor activity during whole word perception. Third, evidence indicates that the motor modeling processes thought to aid in auditory speech perception may be especially engaged during audiovisual speech perception[30–34]. Given that motor regions are more engaged in challenging listening conditions[21,22,35], it is possible that visual speech, which increases the intelligibility of auditory speech, could have variable effects on motor engagement depending on the ambiguity of the overall audiovisual signal[34,36–41]. Fourth, while there is ample evidence to support the idea that the motor system is preferentially engaged when speech input is noisy or ambiguous[12,21,35,42], other results indicate that noise may not be necessary for motor engagement[7,8,28,43,44], and it is unclear how the effects of ambiguity interact with the content of the signal (i.e. auditory words/ phonemes vs. audiovisual words vs. non-speech). Finally, these same motor regions are also involved in more domain-general processes[10,45–47], and most prior studies have been unable to disambiguate between motor modeling and domain-general hypotheses because they used tasks in which participants choose from a small set of speech tokens on every trial[20,27,48]. Small stimulus sets allow for rehearsal or preparation of any perceptuo-motor templates ahead of the upcoming stimulus, meaning that motor modeling can precede stimulus presentation and decision making can begin almost simultaneously with perception. Thus, factors like attention and decision making can potentially influence the degree of motor engagement. The present study addresses these issues in the following ways: (1) comparing perception of speech and non-speech environmental sounds, (2) comparing phonemes and whole words, (3) comparing auditory-only words and audiovisual words, (4) employing adaptive staircase procedures to hold difficulty (ambiguity) constant at two levels (Easy and Hard) and controlling for difficulty across stimulus types, and (5) focusing on perception (as opposed to decision making) by using large sets of stimuli and temporally separating stimulus perception from answer choice presentation and response. To accomplish this, we use a four-alternative forced-choice (4AFC) task and examine EEG signatures of motor activity (μ/beta rhythm event-related spectral perturbations (ERSPs)). Mu/beta (8–30 Hz) power suppression, also known as event-related desynchronization (ERD) and measured as a decrease in EEG power relative to baseline, has been shown to reflect increased motor system activity[27,49–52] including the production of single syllables and words[53].

Using the Rauschecker and Scott dual stream model, we hypothesize that stimulus features and task demands should dynamically determine the recruitment of the dorsal and ventral streams (Fig. 1). Specifically, we hypothesize that perception of relatively unambiguous speech at the word level primarily relies on the ventral stream, without the need for motor involvement. The dorsal stream (motor system) becomes engaged for perception of speech or other auditory signals derived from "doable" actions, contributes to correct perception, and is engaged preferentially under three conditions: (1) when ventral stream processing is insufficient or inefficient because the signal does not contain lexical information, and thus does not match any stored auditory word forms, (2) when ambiguity in the auditory signal creates uncertainty in the selection of a stored auditory word form in the ventral stream, or (3) when the presence of an additional signal or context, such as visual speech, activates motor speech representations. In the first two conditions, ventral stream processors, which rely on mapping the incoming stimulus to auditory templates, are insufficient due to either failure to identify a word (e.g., during perception of phonemes or VCV tokens), or uncertainty in the identity of the word (e.g., under high-noise conditions). Under these conditions, the dorsal stream auditory-motor mapping pathway provides a mechanism to either construct a novel form that matches the auditory signal, or disambiguate the identity of the item. In the third condition, we propose that while motor activity is not necessary for accurate perception, it may be engaged obligatorily in some conditions, for instance when seeing another speaker produce the word.

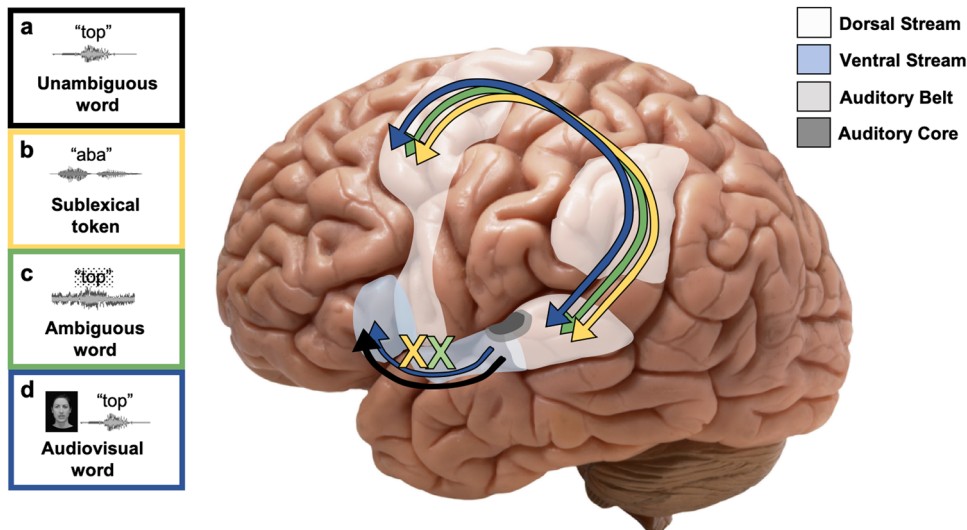

**Fig. 1 Hypothesized dynamic, flexible engagement of the dorsal and ventral streams.** The ventral stream is represented in light blue and the dorsal stream in white. **a** Unambiguous lexical speech relies primarily on the ventral stream (black arrow). Ventral stream processing is insufficient or inefficient for the processing of **b** sublexical tokens (yellow X) and **c** ambiguous/noisy lexical items (green X), and processing these items engages the dorsal stream (yellow/green arrows). **d** While ventral stream processing may be sufficient for audiovisual items, the presence of visual input stimulates obligatory motor modeling and dorsal stream engagement (blue arrows). Brain image taken from open source repository freeimages.com.

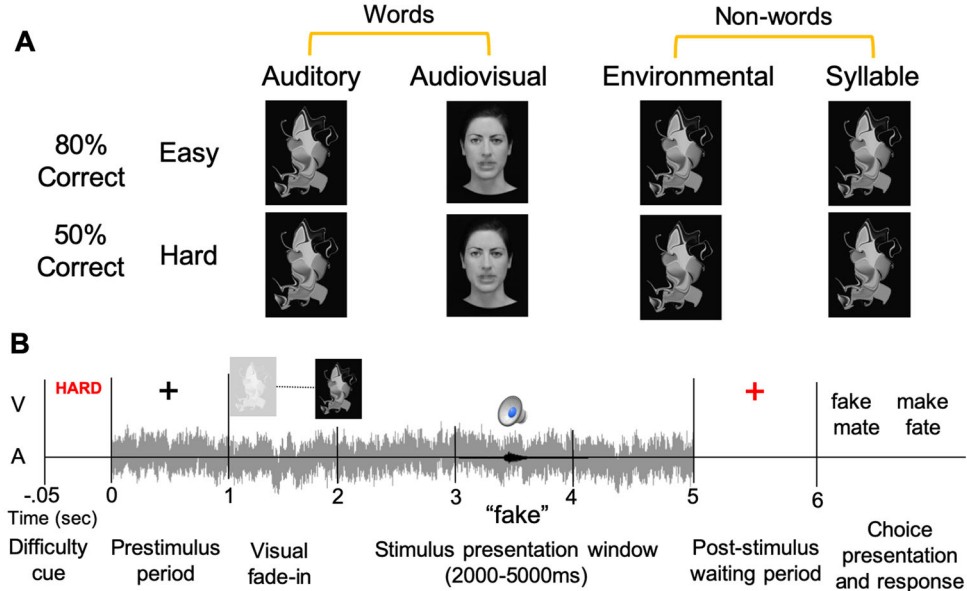

**Fig. 2 Task conditions and trial schematic. A** The eight experimental conditions. The level of noise remains the same, while the intensity of the stimulus is altered to achieve the Easy and Hard SNRs. **B** Trial procedure featuring a sample auditory hard trial. The visual stimulus fades in for 500 ms beginning at 1 s, and remains until offset of the auditory stimulus. Auditory stimulus onset can occur any time after 2000 ms, and offset must occur no later than 5000 ms.

By systematically comparing auditory word perception with the other stimulus types (auditory phonemes, audiovisual words, and meaningful non-speech sounds), controlling for difficulty across conditions, and focusing on a precise stimulus perception widow, the present study demonstrates how specific stimulus features influence the engagement of the motor system during perception. Our results show left hemisphere µ/beta power suppression (enhanced motor activity) during the perception of speech stimuli, but not non-speech environmental sounds. Furthermore, the magnitude of this activity depends on the characteristics of the speech stimulus: audiovisual word and phoneme stimuli showed greater motor activity for correct relative to incorrect trials, while auditory word stimuli showed greater

motor activity for incorrect trials. These findings demonstrate that motor activity is not simply the result of domain-general processes and support a model in which the motor system is flexibly engaged to aid perception depending on the nature of the speech stimuli. Crucially, motor engagement has different effects on the perceptual outcome depending on the lexicality and modality of the speech stimulus.

## Results

**Behavioral results**. The behavioral task is illustrated in Fig. 2. The adaptive staircase procedures controlled for difficulty across stimulus types and maintained performance at roughly 80% and 50% correct for the Easy and Hard levels (Fig. 3). While there

were no significant interactions, the mixed-effects model revealed an effect of difficulty (estimate = −29.09, std. error = 0.63, $p = 0.000$), indicating that the Easy and Hard levels were significantly different within stimulus types. The mean accuracy on the lip-reading task was slightly above chance (28.7% correct; $p = 0.004$, one-sample Wilcoxon sign rank test vs. 25%), well below the AVWord Hard score, demonstrating that participants used both visual and auditory information to perform in the AVWord Hard condition. When assessing differences in volume required to maintain the 80% and 50% correct Easy/Hard thresholds, we found a significant interaction between condition and difficulty ($F (3,69) = 3.75$, $p = 0.015$), as well as a significant main effect of difficulty ($F(1,23) = 1295$, $p < 0.001$) and a significant main effect of condition ($F(3,69) = 60.91$, $p < 0.001$). Post-hoc pairwise $t$-tests (Bonferroni corrected) revealed significant differences in volume between Easy and Hard within each condition, and significant differences in volume between each of the pairs of stimulus types

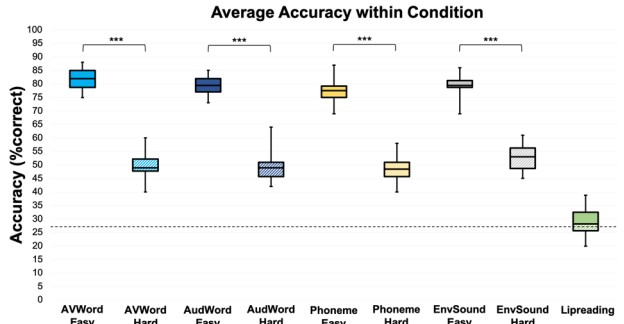

**Fig. 3 Behavioral scores for each of the eight stimulus types and the lipreading task.** All tasks were 4AFC identification tasks. The dotted line represents chance-level performance. Box plots are formatted in the following way: center line is the median; box limits are the upper and lower quartiles; whiskers are 1.5× interquartile range. (Significance levels: * = $p < 0.05$, ** = $p < 0.01$, *** = $p < 0.001$).

with the exception of AVWords and EnvSounds (SI Fig. 1 and SI Table 4). Across both difficulty levels, the AudWord stimuli had the highest mean volume, while the Phoneme stimuli had the lowest. The difference in the volume required to achieve the same accuracy level suggests that the stimulus types differed in their inherent recognizability or difficulty. This further emphasizes the importance of controlling for difficulty levels when comparing behavioral and neural responses across stimulus types.

### EEG results
*Component clustering results.* Independent component (IC) clustering revealed roughly symmetric IC clusters in right and left hemisphere sensorimotor regions (Fig. 4). Total number of ICs per cluster were: left sensorimotor = 1143, right sensorimotor = 1097. As these matched our a priori areas of interest, the analysis was constrained to these sensorimotor clusters.

*Response analysis.* In order to confirm the frequency range of the motor-related ERSPs in this study, we first examined time-frequency activity at the time of the button press (Fig. 5a). We found strong bilateral power suppression in the 8–30 Hz range associated with the button press. Typically, one expects μ/beta suppression related to a motor movement to begin at the time of the response ready signal or "go" cue[54,55], and as expected, we observed suppression in the 8–30 Hz range that began to ramp up before the button press. To confirm that this button press-related μ/beta activity did not contaminate the activity during the stimulus period, we examined activity time-locked to the presentation of response options, which in this experiment serves as the ready cue (Fig. 5b). In the left hemisphere, the μ/beta suppression related to the button press begins after the start of the silent post-stimulus waiting period at ~800 ms prior to answer choice onset, which is well after stimulus presentation.

*Time-frequency results across condition.* Given our hypotheses about motor involvement in perception, our initial analysis examined μ/beta activity across all conditions in the left and right

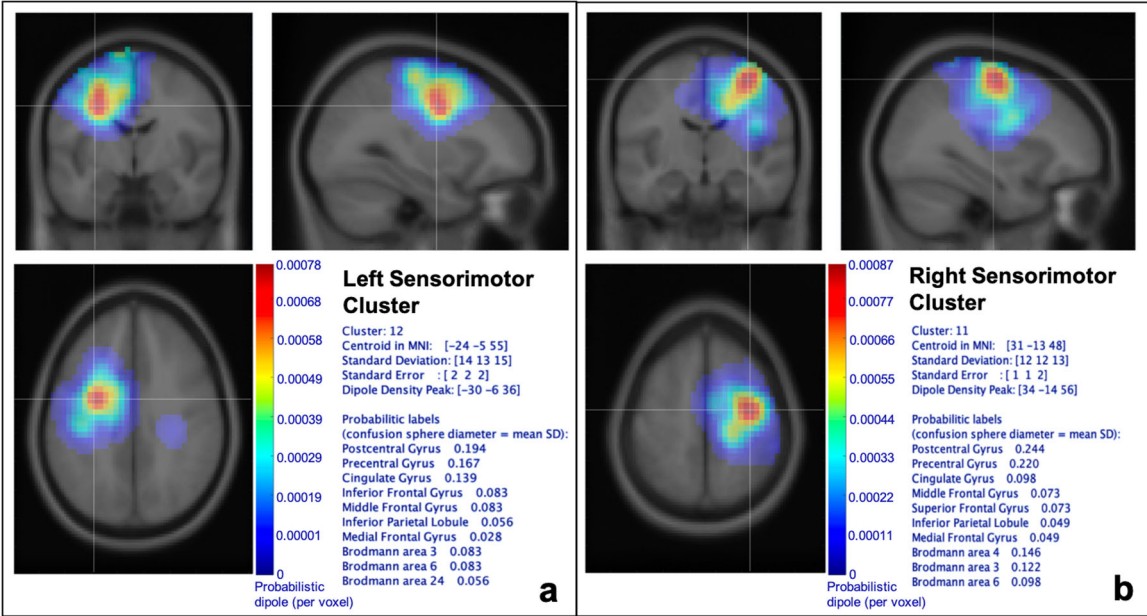

**Fig. 4 Independent component clusters in sensorimotor cortices. a** Left and **b** right sensorimotor clusters. Component clusters were visualized using the EEGLAB plugin *std_pop_dipoledensity*, with a Gaussian smoothing kernel set to FWHM = 15 mm. The probabilistic labels are determined using a confusion sphere centered on the dipole density peak. The diameter of the sphere is the mean of the standard deviation of the centroid in MNI space, and the labels represent the regions with the highest dipole concentration in the cluster, listed first by anatomical label and then by Brodmann area.

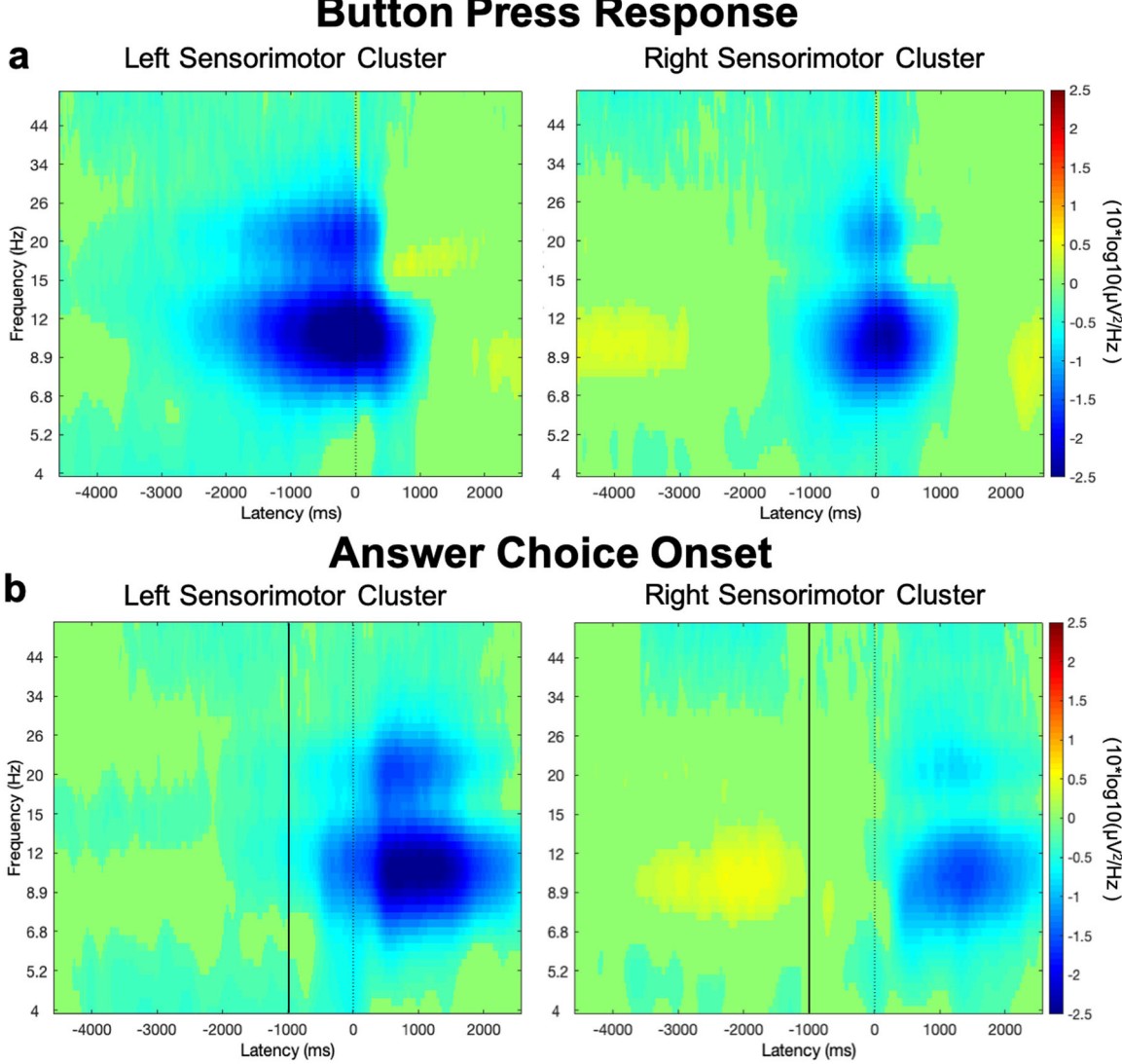

**Fig. 5 EEG activity during motor response.** In both hemispheres, the strongest response-associated activity appears within the μ and beta frequency ranges (8–30 Hz). All plots are baseline corrected at $p < 0.001$, FDR corrected. **a** EEG activity time-locked to the button press (dotted line). The button press occurs at variable latencies, so previous trial events (e.g. answer onset) are smeared across the pre-response period. **b** EEG activity time-locked to the onset of the answer choices (dotted line). The start of the silent post-stimulus waiting period occurs at −1000 ms (solid line). The stimulus presentation window stretches from −4000 ms to −1000 ms, and stimulus onset times vary across this window. In the left hemisphere, motor activity begins to ramp up −500 ms prior to answer choice onset. In the right hemisphere, motor activity does not begin until after answer choice onset.

sensorimotor clusters in the 8–30 Hz window identified in the button press analysis. We found a significant difference in μ/beta activity between the hemispheres (estimated mean difference = −0.33, std. error = 0.04, $p < 0.001$). Within the time-frequency analysis window (−200 to 1000 ms relative to stimulus onset and from 8–30 Hz), mean EEG activity in the left sensorimotor cluster (collapsed across all conditions) showed strong μ/beta suppression relative to baseline that began prior to stimulus onset, which continued through the epoch window (estimate = −0.13, std. error = 0.06, $p = 0.048$, Fig. 6). In contrast, the right sensorimotor cluster showed no evidence of suppression, and instead showed increased power (event-related synchronization, ERS) in the 8–30 Hz range (estimate = 0.20, std. error = 0.06, $p = 0.002$; Fig. 6). Given that μ/beta suppression is indicative of increased motor activity[27,49,50,53,56], this result suggests that motor activity during perception is left-lateralized. As such, the main analyses focus on the left sensorimotor cluster.

*Sensorimotor time-frequency results by stimulus type.* Within the left sensorimotor cluster, we examined effects of stimulus type and accuracy on motor activity indexed via μ/beta suppression, with greater suppression signifying greater motor activity[27,49,50,53,56]. First looking within stimulus type, we find a pattern of enhanced μ/beta suppression for correct relative to incorrect trials for the AVWords (estimated difference = 0.29, std. error = 0.11, $p = 0.01$), as well as for Phonemes, although for Phonemes the difference was not significant (estimated difference = 0.16, std. error = 0.10, $p = 0.10$; Fig. 7a). In contrast, the AudWord stimuli showed the opposite pattern: enhanced suppression for incorrect trials relative to correct trials (estimated difference = −0.33, std. error = 0.11, $p = 0.003$). EnvSounds did not show a significant difference between incorrect and correct trials and the estimated difference was comparatively small (estimated difference = −0.02, std. error = 0.10, $p = 0.80$). These differences between correct and incorrect trials within each stimulus type highlight the significant interaction between stimulus

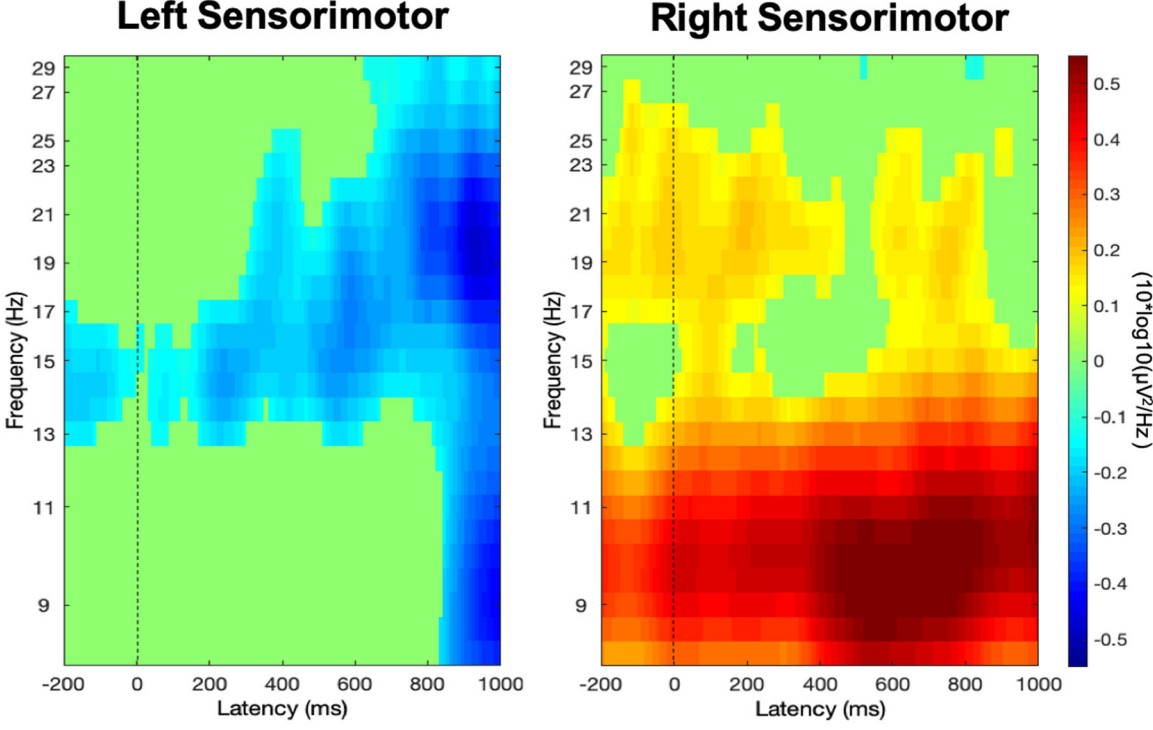

**Fig. 6 Stimulus-locked EEG activity across all conditions.** EEG activity across all conditions in the left and right sensorimotor clusters in the pre- and peristimulus time window, masked for significant differences from the baseline period at $p < 0.05$, FDR corrected. Stimulus onset occurs at time zero (dotted line).

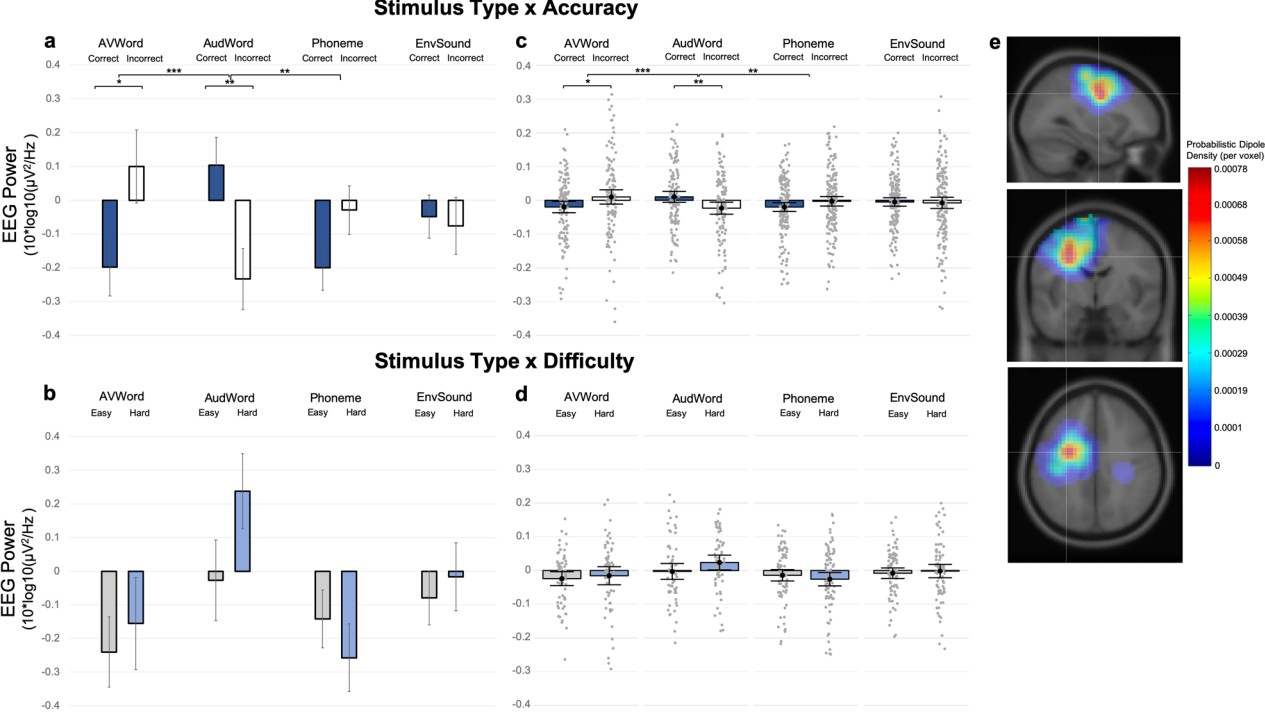

**Fig. 7 EEG activity in the left sensorimotor cluster.** Mean EEG power ($10 \times \log10(\mu V^2/Hz)$ for **a** stimulus type and accuracy and **b** stimulus type and difficulty level. Plots in **c** and **d** show the same mean values as **a** and **b**, but with individual data points overlaid. **e** Left hemisphere sensorimotor cluster location, created using EEGLAB plugin *std_dipoleDensity()*, with a Gaussian smoothing kernel set to FWHM = 15 mm. Error bars represent mean ± 1.96 × standard error. Significance levels: * = $p < 0.05$, ** = $p < 0.01$, *** = $p < 0.001$. Our analysis used a series of mixed-effects models with subject and independent component (IC) as the random effects. The $n$ for the above plots differs by stimulus type and accuracy/difficulty and is given in SI Table 5. The average number of ICs per subject per condition is given in SI Table 6.

type and accuracy ($\chi^2(3) = 17.6$, $p < 0.001$) found in the mixed-effects model. This interaction was driven by the differences between the AudWord and AVWord stimuli (estimate = 0.62, std. error = 0.16, $p < 0.001$) and between the AudWord and Phoneme stimuli (estimate = 0.50, std. error = 0.15, $p = 0.006$). The difference between correct and incorrect trials was greater for the AudWords than for the EnvSounds, although this difference did not reach statistical significance after Bonferroni correction (estimate = 0.31, std. error = 0.15, $p = 0.22$).

Next, we examined effects of stimulus type and difficulty level (using correct trials only) and did not find a significant stimulus type by difficulty interaction ($\chi^2(3) = 5.0$, $p < 0.17$). Descriptively, however, we see that the difference between Easy and Hard trials was greater for the AudWords than the Phonemes or the AVWords (Fig. 7b). While the lack of a significant interaction means that any main effects need to be interpreted with caution, we ran a separate model to investigate whether we saw the same pattern of responses for the different stimulus types when looking at the Easy/Hard correct trials. We found a significant main effect of stimulus type ($F = 4.27$, $p = 0.006$). The pattern of μ/beta suppression showed greater motor activity regardless of difficulty for AVWord and Phoneme stimuli than for the AudWord stimuli (AudWord vs AVWord, estimate = −0.29, std. error = 0.10, $p = 0.02$; AudWord vs Phoneme, estimate = −0.29, std. error = 0.11, $p = 0.04$). The AudWord stimuli showed almost no μ/beta suppression for Easy trials and increased μ/beta power for Hard trials (Fig. 7b). Again, there was very little modulation of μ/beta activity for the EnvSound stimuli.

Given that the observed μ/beta ERSPs for the EnvSounds did not appear to cross the zero axis (thus indicating no reliable difference from baseline activity) in either the accuracy or difficulty analyses, we used one-sample tests to investigate whether any activity in the EnvSound condition was significantly different from baseline. Mu/beta suppression was not significantly different from baseline in any of the four EnvSound conditions (EnvSound Correct, $M = −0.049$, SD = 0.811, $p = 0.448$; EnvSound Incorrect, $M = −0.076$, SD = 1.086, $p = 0.370$; EnvSound Easy, $M = −0.079$, SD = 0.722, $p = 0.322$; EnvSound Hard, $M = −0.017$, SD = 0.898, $p = 0.870$, uncorrected). Each of the other stimulus types showed significant modulation in one or more of the conditions (Aud-Word Incorrect, $M = −0.233$, SD = 1.011, $p = 0.039$; AudWord Hard, $M = 0.238$, SD = 0.863, $p = 0.037$; AVWord Easy, $M = −0.241$, SD = 0.810, $p = 0.025$; Phoneme Correct, $M = p = 0.019$; Phoneme Hard, $p = 0.013$, uncorrected; SI Table 3).

## Discussion

Our analysis of μ/beta suppression in sensorimotor regions suggests that motor involvement in perception is left-lateralized. While the left hemisphere sensorimotor cluster showed increased motor activity in response to the speech conditions, we observed no significant modulation of motor activity to non-speech environmental sounds, suggesting that this motor activity during perception is not related to domain-general processing but rather is specific to speech. Furthermore, we found that motor engagement relates to correct perception of phonemes and audiovisual words but incorrect perception of auditory-only words. In line with our hypotheses, these findings suggest that the motor system is flexibly engaged to aid perception depending on the nature of the speech stimuli. More specifically, they suggest that while motor activity aids in the perception of phonemes and audiovisual words, processing auditory-only words via this motor mechanism is ineffective. Other findings did not support our hypothesis: we did not observe increasing motor engagement with increasing ambiguity, and motor activity was unexpectedly related to incorrect perception of auditory-only words.

In this study, our measure of motor activity was μ/beta suppression (8–30 Hz) in the left sensorimotor cluster. Previous literature has demonstrated that in sensorimotor areas, suppression in this 8–30 Hz range is related to increased motor activity[49,56–58], and this relationship between μ/beta suppression and motor action has also been observed in EEG investigations of speech perception and production[27,53,59]. However, it is important to note that μ/beta suppression (ERD) has also been associated with cognitive functions like the anticipation of an upcoming response[46,60] and cued attention[61,62]. While the functional roles of these oscillations are still a matter of debate, several factors indicate that the sensorimotor μ/beta suppression observed here is specifically related to speech perception and cannot be explained by anticipation, attention, other domain-general or non-perceptual processes. First, all stimulus types were presented within the same trial structure, meaning that the timing (and thus anticipation) across types was held constant. Given that the task structure and difficulty levels were constant across stimulus types, the different patterns of results, both within the speech stimuli and between the speech and environmental sound stimuli, suggest that the observed μ/beta suppression is not simply a reflection of general task demands such as attention, decision making, cognitive control, or motor preparation for a button-press. Second, our task was specifically designed to limit the conflation of attentional processes with speech perception in several ways: encouraging identification of the stimulus at the time of presentation, separating perception from decision making and motor preparation for the button press, and using a large stimulus set to preclude the pre-loading of perceptual-motor templates of the response options. Third, the temporal resolution of EEG and the restriction of our analysis to the pre- and peri-stimulus period indicates that the observed μ/beta suppression is likely not a reflection of verbal working memory or early covert production, which are common criticisms of studies investigating these mechanisms on an extended time scale (refs. [20,63]). Finally, the pattern of μ/beta suppression observed in our study closely matches the μ/beta ERD measured from a similar left hemisphere cluster in a previous investigation of speech production by Jenson and colleagues[53], suggesting that the μ/beta suppression reported here most likely reflects motor activity.

Our results indicate that increased activity in left hemisphere motor regions may support the processing of specific types of speech stimuli, possibly through internal modeling of speech representations or a similar phoneme-level perception mechanism. Furthermore, these results suggest an interactive network for speech perception in which the dorsal and ventral streams are flexibly engaged, and indicate that the degree to which the two streams are engaged has different effects on perceptual accuracy depending on stimulus type. Below we outline a number of pieces of evidence that support this claim.

The strongest support for the idea that motor activity reflects speech perception processes, and in particular, the modeling of internal representations, comes from the observed differences between the speech and environmental sound stimuli. Each of the three speech stimulus types was accompanied by modulations in μ/beta activity in the left hemisphere sensorimotor cluster. In contrast, there was no significant modulation of μ/beta activity for the environmental sounds, which aligns with prior literature showing that non-speech sounds like tone-sweeps do not significantly engage motor areas during perception[27,64]. Our results demonstrate that in contrast to speech stimuli (both words with semantic content and sub-lexical phonemes), the motor system is not significantly modulated by semantically relevant, complex sounds, thus providing further evidence for the hypothesis that motor activity during perception is specific to speech and may represent the modeling of articulatory representations.

The strong left-lateralization of the observed μ/beta suppression provides further support for the hypothesis that this activity relates to speech perception. Both the time course of the μ/beta activity and the differences between conditions suggest that this lateralization does not simply reflect preparation for a right-handed response. First, during stimulus perception the left and right hemispheres show opposite patterns of μ/beta activity (Fig. 6), while during the button press there is strong bilateral μ/beta suppression (Fig. 5) as is typical during a motor response[65]. In our task, the button press did not occur until more than 1 s after stimulus offset, and we found that, consistent with prior literature, response-related increases in μ/beta suppression in both hemispheres began at answer choice onset, and were temporally centered around response onset[54,55]. Second, the button press was constant across conditions, and it is unlikely that motor response preparation could account for the different patterns of μ/beta activity during perception of the different stimulus types. Thus, our data suggest that during perception there is a functional specialization of motor activity between hemispheres. Prior literature has shown that speech processing is strongly left-lateralized, especially in dorsal stream regions[3,4,66]. In addition to neuroimaging experiments showing a strong leftward-bias during speech processing[67], lesion studies demonstrate that unilateral left frontal strokes are accompanied by lasting severe speech production deficits[68–71]. We suggest that the left–right division in μ/beta activity provides additional support that the observed left hemisphere motor processing is specific to speech.

In contrast to the μ/beta suppression in the left hemisphere, the right hemisphere exhibited a pattern of increased power centered on the μ frequency. Event-related increases in alpha power have been shown to reflect inhibition of task-irrelevant or distracting information[72,73], and both alpha/μ and beta rhythms have been described as "idling rhythms" that display greater power at rest and in which power suppression indicates release from inhibition[56,72]. Research suggests that inhibitory processing is right-lateralized[74], and increased μ/beta power in right hemisphere frontal regions has been related to inhibition of irrelevant stimuli in a tactile detection task[75]. Most closely related to the current study, Strauß et al. demonstrated that increased alpha power may reflect auditory selective inhibition of noise during listening to noise-masked speech[76]. Given that we find increased μ power in the right hemisphere across all conditions, our results are consistent with the notion that the right hemisphere may help suppress task-irrelevant information.

Returning to the left hemisphere and examining the speech perception conditions, we found opposite patterns of motor activity for the AVWord and Phoneme conditions relative to the AudWord condition. For both audiovisual words and auditory-only phonemes, greater motor activity during perception was related to correct perception. This result aligns with our hypothesized role for the motor system in processing sublexical and audiovisual stimuli. Taken in the context of prior literature showing that motor regions display category-specific responses during phoneme perception[35,77,78], our results support the idea that motor activity reflects the modeling of internal phoneme representations. The fact that this activity is greater for correct trials supports the idea that these representations help to decode sublexical speech input. Importantly, the results of this study replicate findings from several prior studies showing that motor activity relates to correct phoneme perception in noise[27,35,53], and extends these findings by showing that this same pattern of activity is not present during the perception of auditory-only words. In fact, we show that greater motor activity (μ/beta power suppression) is related to incorrect perception of auditory-only words, and that correct perception is instead related to increased μ/beta power. As mentioned above, increased power in the μ/beta band has been shown to reflect inhibitory processing[72,73,76]. Therefore, our results suggest that relying on the motor system to process lexical stimuli is ineffective, and perhaps that inhibiting motor system processing may even be helpful when perceiving auditory-only words. Alternatively, it could be that, as hypothesized, ventral route processing is normally sufficient for auditory-only words, but during these incorrect auditory-only word trials, ventral route processing is insufficient and the dorsal stream is recruited in an unsuccessful effort to compensate and salvage perception. This could explain the greater motor engagement seen for incorrect auditory-only words and the lack of motor activity for correct auditory-only words. By either interpretation, motor activity does not contribute to successful perception of auditory words.

Despite this finding for auditory-only lexical items, we find that when lexical stimuli are accompanied by the corresponding visual speech stimulus (talking face), we again see that increased motor activity is related to correct perception. This indicates that while engaging the motor system for auditory-only lexical stimuli may be ineffective, these motor mechanisms aid in the perception of audiovisual lexical stimuli. Visual speech has been shown to engage the same frontal motor regions active during the production or silent rehearsal of speech[30,34,79]. Given the shared pattern of activity for the AVWords and Phonemes, perhaps the presence of visual speech information biases processing toward or facilitates access to sublexical motor modeling that can further aid the decoding of auditory input during audiovisual perception. Prior literature demonstrates that the presence of visual speech makes auditory speech easier to understand by providing additional information to constrain the interpretation of auditory input[36], and our results suggest that the motor system provides the neural basis for this process.

Why would this motor mechanism be ineffective without the presence of visual speech information? Previous research indicates that auditory lexical stimuli are processed rather efficiently in the ventral stream[14], and that the presence of visual speech can speed neural responses to auditory speech[32,80]. Perhaps during auditory-only lexical perception, motor-generated sublexical template representations are less efficient than relying on ventral stream processing alone, but the addition of visual speech allows for earlier access to these templates and therefore a more beneficial use of the modeling mechanism. It is worth noting that we do not find a difference in the magnitude of motor activity during the perception of phonemes and audiovisual words. It is possible that both motor system processing of phonemes and audiovisual words involves the same perceptuo-motor models, or that these two stimulus types engage separate cognitive processes within motor areas. Future studies using multivariate classification methods may be able to address this question. More detailed information regarding the onset and duration of the μ/beta suppression for the different stimulus types would also be useful; however, the variability of the stimuli in this study makes it poorly suited to assessing differences in temporal characteristics between stimulus types. Thus, further investigation is needed to more fully characterize the different neural processes underlying the perception of auditory words, audiovisual words, and phonemes and to better understand why motor activity appears to be beneficial for some stimulus types but not others.

As mentioned above, there is ample support for the idea that motor activity may be preferentially engaged when the stimuli are noisy or degraded[16,17,21,23,35,48], however, our data do not support this claim. It may be that in our task, the Easy condition already contained sufficient noise to evoke motor activity, and that this lowered our ability to detect differences between Easy and Hard trials. A second possibility is that there is a non-linear relationship between difficulty and motor activity such that too

little ambiguity does not evoke motor activity, but too much makes the signal unintelligible and also results in less motor activity. If this is the case, perhaps our Easy and Hard conditions fell at points on this inverted u-shaped curve that did not allow us to detect effects of difficulty. Finally, almost all the prior studies investigating the effects of ambiguity on motor activity have used limited stimulus sets, which may allow participants to prepare motor templates of the response options prior to stimulus presentation, thus conflating perception with decision-related processing. If the previously observed relationship between motor activity and ambiguity was related to decision-making processes, perhaps by addressing this confound in our study we removed the effect of difficulty, thus focusing on an independent, perception-specific aspect of motor processing.

While the results of the current study provide evidence that left hemisphere motor regions may be beneficial for processing for certain types of speech stimuli, there are several important limitations to consider. First, the data presented here are correlational, and do not reflect causal evidence. It is worth noting that a prior study found that disruption of PMC impaired phoneme perception but not word-level perception, lending causal support for the dissociation between auditory words and phonemes reported here[29]. However, in order to understand the exact mechanisms and their functional relevance, more studies involving causal methods are needed. The second limitation of this research is the spatial resolution of EEG source localization techniques. Even with the most precise EEG source localization techniques, the spatial resolution is such that we cannot separate the relative contribution of nearby motor areas (e.g. ventral PMC and M1). Future investigations of this question would benefit from combining techniques like EEG and fMRI to resolve both the time course and exact spatial locations of these processes within cortical motor regions.

In conclusion, motor processing appears to be selectively related to the perception of speech stimuli and is not simply a result of domain-general processes like attention or decision making. Furthermore, we demonstrate that activity in left hemisphere motor regions aids in the correct perception of auditory phonemes and audiovisual words, but is ineffective for processing auditory-only words. These results provide evidence for an interactive network for speech perception in which dorsal stream motor areas are dynamically engaged during the perception of speech stimuli. Crucially, this motor engagement has different effects on perceptual outcome depending on the type of stimulus.

## Methods

**Participants**. Participants were 24 healthy, right-handed adults (mean 23.9 yrs, range = 21–31 yrs, 16 females) with no history of neurologic, psychiatric, or learning disorder, normal hearing, normal or corrected to normal vision. Handedness was assessed using the Edinburgh Handedness inventory[81], and all participants were native English speakers. Participants were recruited from Georgetown University and the surrounding community, provided informed consent, and were compensated as approved by the Georgetown University Institutional Review Board. While 34 participants completed both sessions of the experiment, one was excluded due to late age of English acquisition, two due to equipment errors, two due to not having the minimum number of ICs (see SI Note 4) and five due to overall noisy data. In addition to these local participants, an additional 288 participants were recruited for a stimulus norming procedure conducted through Amazon Mechanical Turk[82]. Online participants were also provided informed consent and compensated in accordance with the Georgetown IRB.

**Stimuli**. We used a 4AFC task in which a stimulus was presented, and then following a delay, the four options appeared as written words. We created four types of stimuli: auditory-only words (AudWords), audiovisual words (AVWords), auditory-only phonemes (Phonemes), and non-speech, auditory-only environmental sounds (EnvSounds). Stimulus types were constructed so that each differed from the AudWords by one feature: AVWords differed in the addition of visual input, Phonemes differed in the lexical status of the stimuli, and EnvSounds were semantically meaningful but were not speech. The AudWord condition served as the reference point for comparison with the other conditions. All stimuli had a

mean duration of 1139 ms (SD = 239 ms, range 499–1935 ms). See SI Note 1, Stimulus Details for further details on the content of each stimulus category. A full list of the words, phonemes, and environmental sounds used in each condition of the experiment can be found in Supplementary Data 1.

**Task and procedure**. The behavioral task was administered using a custom script in the PsychoPy platform[83]. In order to manipulate the difficulty level of the task, the auditory file for each stimulus was masked using pink noise and SNR was manipulated by changing the volume of the stimulus, such that the level of noise remained constant in all trials. Because the noise remained constant, and thus the first 1.5 s of each trial were identical, a difficulty cue was presented at the beginning of each trial to set the participant's expectation and allow for effects in the prestimulus period (Fig. 2). Participants were seated in a dark room and instructed to keep their eyes open and fixated at the center of the screen for the duration of the experiment. They were told to listen carefully and select the answer that best matched what they heard. Reponses were made by using the right hand to press one of the four arrow keys on a keyboard. We choose to control for response-related activity by separating the response from stimulus presentation by a minimum of 1000 ms, and we performed a separate analysis (see Methods section "Statistical analysis of EEG data") to ensure that response-related motor activity did not contaminate the stimulus perception window. Stimulus conditions were presented in five blocks of 20 trials each, with the Easy/Hard trials alternating within a block. Thus, each participant completed 100 trials per stimulus type/difficulty combination, for a total of 800 trials in the entire experiment, which was split across two sessions (SI Fig. 2). The AVWord and AudWord conditions were presented in one session, with the Phoneme and EnvSound conditions in a separate session. Sessions were separated by at least 1 day and no more than 2 weeks. Each trial began with the difficulty cue, followed by the onset of the pink noise and a 1000-ms prestimulus period in which participants heard only the noise and viewed a fixation cross. Following the prestimulus period, visual stimulation (morphed image or still frame of the video) was faded in for all conditions. Stimulus onset was jittered, and could occur within a 3000-ms window. In the AVWord condition, the still frame of the video began moving with stimulus onset. Response options were shown after a 1000-ms waiting period in order to ensure that decision making and button press response preparation were separated from stimulus identification. The delay also encouraged identification of the item rather than reliance on echoic memory during the response period, and prevented the preparation of a perceptual-motor model of the response options prior to stimulus onset. The task used adaptive staircase procedures (PsychoPy MultiStairHandler) to achieve an overall accuracy of 80% in each of the four Easy conditions and 50% in each of the four Hard conditions. We chose to present the Easy stimuli in noise to avoid ceiling effects and match input in the prestimulus period. Also note that for a 4AFC task, 80% is still quite easy and 50% is above chance. For more information on stimulus recording and preparation, see SI Note 2.

**EEG recording**. EEG data were collected using a 128-channel HydroCel Geodesic sensor net (Electrical Geodesic, Inc., Eugene, OR) and digitized at a sampling rate of 500 Hz. Halfway through each experiment session, the behavioral task was paused and the sensor impedances were checked and adjusted if necessary such that impedances were kept below 70 kΩ for the duration of the experiment. All channels were referenced to the vertex, and a bandpass hardware filter (0.1–100 Hz) was applied. All other EEG preprocessing was performed offline.

**Behavioral analysis**. All analyses were conducted in R Studio[84]. To assess differences in accuracy between conditions, we conducted a 4 × 2 mixed-effects model analysis using the *lmer* package with stimulus type and difficulty as fixed effects and subject as a random effect (*model = lmer (Score ~ Condition + Difficulty + (1| SubID), data = Data*). We also tested for correlations between each stimulus type/difficulty level as well as correlations with the lipreading scores. Finally, since the adaptive staircase procedure (SI Note 3) set a custom volume for each stimulus type/difficulty level for each participant, we assess differences in the average volume (dB) using a 4 × 2 repeated measures ANOVA using the *anova_test* package (*res.aov <- anova_test(data = Data, dv = Volume, wid = SubID, within = c(Difficulty, Condition))*).

**EEG analysis**

*Preprocessing*. EEG analyses were conducted in EEGLAB[85]. The data were first low-pass filtered (FIR, Hamming window, filter order 330, cutoff frequency 57.5 Hz), then down-sampled to 250 Hz, and finally high-pass filtered (FIR, Hamming window, filter order 660, cutoff frequency 2 Hz). The data were subjected to a combination of manual and automatic cleaning and artifact rejection procedures, including adaptive mixture independent component analysis (AMICA), which identifies stationary brain and non-brain (i.e. artifactual) source activities (see SI Note 4, EEG Preprocessing). Following data cleaning and rejection, continuous EEG data were epoched from −2 to 1.5 s relative to the onset of the stimulus sound and subjected to a final round of quality control measures in channel space, and the final average number of trials per condition per subject is shown in SI Table 2. Performance was held at 80% correct for the Easy level, meaning the Easy Incorrect category necessarily had a small number of trials. Given this low trial number,

statistical analysis of the EEG data examined the effect of accuracy by collapsing across difficulty, thereby ensuring a sufficient number of trials in each condition (Correct (Easy + Hard) vs. Incorrect (Easy + Hard)). See section "Statistical analysis of EEG data" below for more detail on statistical analyses. Equivalent current dipoles were estimated for each IC, and going forward all analyses were performed at the IC (source) level rather than on channel data.

*Time-frequency analysis and component clustering.* All datasets were again loaded into the EEGLAB STUDY structure, and time-frequency analysis (group-level ERSP) was performed using a blend of Morlet-based Wavelet Transform (WT) and short-term Fourier transform (STFT). An 836 ms sliding window was used to generate time-frequency data from −1 to 1.46 s (10 ms step) relative to stimulus sound onset and from 4 to 55 Hz (log-scaled, 50 bins) while linearly increasing the number of cycles from 3 cycles at the lowest frequency (4 Hz) to 10 cycles at the highest (55 Hz). Single-trial values were averaged across trials then converted to decibel (dB) units by dividing the by the mean baseline period power for each frequency bin. This process automatically achieves cross-frequency normalization, which addresses the issue arising from blending WT and STFT. The baseline period was the second half of the prestimulus noise period, before visual stimulus fade-in (418 to 582 ms relative to the onset of the noise, Fig. 2). This period contains identical stimulation during all conditions, and using this baseline removes the effect of listening to noise that is common to all conditions. Given our a priori hypotheses regarding motor activity during perception, we chose to examine the time-frequency activity at the source level. To do so, we used the EEGLAB STUDY framework to perform *k*-means clustering based on dipole location only (SI Note 5, Component Clustering). Time-frequency results within a cluster of interest were compared across conditions. While there are general limitations of EEG source localization that also apply to our methods, including the use of template channel location, use of template MNI brain, and imperfect knowledge about electric conductivity parameters in the forward model, dipole approximation is known to work well for the EEG sources closer to the surface[86]. Because our region of interest includes M1 and premotor areas, the localization error was expected to be minimal.

*Statistical analysis of EEG data.* We tested the effects of both accuracy and difficulty level on our measure of motor engagement, μ/beta power suppression. Power suppression, or ERD, in the μ/beta band has been shown to reflect increased motor system activity[49,50]. Previous literature has focused on a range from ~8–30 Hz for investigations of motor-related EEG oscillations[27,51,52], including the production of single syllables and words[53]. To confirm that this frequency range was related to motor activity in our participants, we first examined time-frequency activity across all conditions during the button press response. Since all responses were made using a right-handed button press, there was some concern that motor activity related to the button press may contaminate the stimulus perception period; therefore, we also examined activity locked to the onset of the answer choices before proceeding to the main analyses on the stimulus presentation period.

To focus the analysis on stimulus-related processing and limit the potential influence from post-perceptual subvocal rehearsal of the stimulus, ERSP data from 8–30 Hz and from −200 to 1000 ms relative to stimulus onset were submitted for final analysis. Previous work has shown that prestimulus EEG activity is related to perceptual performance, with alpha (or μ) power suppression indicating a release from inhibition and preparation for an upcoming stimulus[87,88], and prestimulus μ/beta suppression has been shown for correct trials during phoneme perception, perhaps signaling a ramping up of motor processing[20]. Thus, we began our analysis window 200 ms prior to stimulus onset in order to incorporate preparatory activity. Within a given cluster, the mean EEG power within this time-frequency window was obtained for each IC for all trials within a stimulus/difficulty condition in that cluster. These ERSP values were then entered into three linear mixed-effects models using the *lmer* package (R Studio)[84]. First, we tested for differences between the left and right hemisphere sensorimotor component clusters (collapsed across conditions) with cluster as the fixed effect and subject and IC as random effects. Next, within the left hemisphere motor cluster, the time-frequency values were entered into two 4 × 2 mixed-effects models with subject and IC as random effects and either (1) stimulus type and accuracy as fixed effects, or (2) stimulus type and difficulty as fixed effects. Following each model, a likelihood ratio test was used to assess whether the interaction terms significantly improved model fit. The equations for the two mixed models in R are as follows (where IC refers to independent component):

Model 1: Accuracy

$$model1 = lmer(EEGmeans \sim Condition * Accuracy + (1|Subject) + (1|IC\_Names), data = Data)$$

Model 2: Difficulty

$$model2 = lmer(Windowmeans \sim Condition * Difficulty + (1|Subject) + (1|IC\_Names), data = DataCorrect)$$

The total number of ICs per condition in the left sensorimotor cluster are shown in SI Table 5, and the average number of ICs per subject per condition are shown in SI Table 6.

Since correct trials represent those during which the participant is engaged in the task and accurately perceiving the stimulus, the effects of difficulty level (Easy vs. Hard) on EEG response were examined using correct trials only. Due to the low number of trials in the Easy Incorrect category, the effects of accuracy (Correct vs. Incorrect) were assessed by collapsing across difficulty level. Before the mixed-

effects model analysis, an outlier analysis (following Tukey's rule of outliers) was performed within each stimulus type/difficulty level combination and values that were below Q1 − 1.5IQR or above Q3 + 1.5IQR were removed[89]. In addition to the 4 × 2 mixed model analyses, we also evaluated whether each stimulus type/difficulty and stimulus type/accuracy result was significantly different from baseline using a one-sample *t*-test (one-sample Wilcoxon signed rank test for non-normal data).

**Statistics and reproducibility**. The statistical tests and software packages used for each analysis are described in detail in the corresponding Methods and Results sections. As described in the Methods section "Participants", all behavioral and EEG analyses included data from 24 participants. For the EEG analyses, we used an IC clustering procedure to extract EEG activity from the left and right hemisphere sensorimotor regions. Different subjects contributed a different numbers of ICs (SI Table 6), which we accounted for by using a linear mixed-effects models with subject and IC as random effects. The total number of ICs per condition for the linear mixed-effects models are given in SI Table 5.

**Reporting summary**. Further information on research design is available in the Nature Research Reporting Summary linked to this article.

## Data availability

See Supplementary Data 2–4 for data underlying the following figures: Fig. 3 (Supplementary Data 2), Fig. 7 (Supplementary Data 3), and SI Fig. 1 (Supplementary Data 4). The remaining EEG data that support the findings of this study are available from the corresponding author upon reasonable request.

## Code availability

The custom code used to create our behavioral task is available at Zenodo[90]: https://doi.org/10.5281/zenodo.4279601.

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

## Acknowledgements
This work was supported by a National Science Foundation (NSF) Graduate Research Fellowship (#1444316 to author K.M.). The authors wish to extend their sincere gratitude to Kathryn Schuler and Andrew Demarco, who were instrumental in programming the behavioral task. We thank Guinevere Eden, Max Riesenhuber, Josef Rauschecker, and Virginie van Wassenhove for their valuable feedback on this study. Finally, we thank everyone in the Cognitive Recovery Lab for their help and support throughout this project.

## Author contributions
K.M., A.M.M., and P.E.T. designed the experiment. K.M. recorded and normalized the stimuli, created the behavioral task code, and collected the data with supervision from P.E.T. Authors M.M., K.M., and G.N. performed the data analysis, with input from A.M.M. Authors K.M. and P.E.T. wrote the manuscript, with input from M.M. and G.N. All authors assisted with data analysis and manuscript revision and approved the final manuscript.

## Competing interests
The authors declare no competing interests.
