## [Peer Review File · Communications Biology]

Reviewers' comments:

Reviewer #1 (Remarks to the Author):

The authors investigated putative motor involvement in speech encoding across multiple stimulus conditions to probe whether the dorsal speech processing route's recruitment varies based on stimulus type and difficulty. They used extra-cranial EEG during a forced-choice identification task to isolate two EEG clusters, localized to the best of our abilities to the left and right sensory-motor areas. These clusters demonstrated strong modulation in the 8-30Hz range by stimulus type. First, the left hemisphere was found to have larger decreases in power relative to the right (with prior literature tying decreases in 8-30Hz power to increased motor activity). In this left cluster they found that phoneme and audiovisual speech but not word speech to be associated with decreased power on correct trials. The authors interpret these results to suggest a left-lateralized dorsal speech perception route which is dynamically engaged based on stimulus properties.

Overall, I found this to be a well-conducted study with an intricately designed behavioral task and well-defined EEG analyses. I believe the question regarding motor involvement in speech perception is important to our overall understanding of language. However, before recommending the paper for publication I have some (mostly minor) comments:

- 1) The last analysis in 2.2.4 regarding the significance of power differences from baseline raises an interesting question. It appears that no condition across stimulus types would differ significantly from baseline after correction for multiple comparisons based on the p-values reported (all $p > .01$ with a high number of one-sample tests run). Does this surprise the authors? Does this suggest to the authors that the motor activity engaged during language perception is at a much lower power than would be engaged by overt action? It seems to make sense that the engagement of speech templates at a sub-movement level for speech perception would evoke much less activity than speech production even if generally the same neural populations are involved. You mention a previous study, [53], which focused on speech production using a similar paradigm. A comparison of overall magnitudes between production and perception would be of interest.
- 2) It may be worth highlighting the lack of difference in 8-30Hz power between audio-visual words and phonemes in the broader discussion. Presumably the neural encoding of visual speech stimuli and the encoding of phonemic auditory speech stimuli are two different processes at circuit level. However, you demonstrate that in sensory-motor areas an overall similar amount of motor-related activity is evoked during processing of both stimuli. Do the authors believe the cognitive operations being performed in the audiovisual and phoneme trials are similar or do these trials just evoke similar amounts of overall neural activity?
- 3) Building off of #2, is it possible to get a temporal marker of the onset of significant 8-30Hz decreases from baseline for each stimulus type post stimulus presentation? Any temporal information would add additional constraints to interpretations as to how processing these different stimulus types proceeds in the sensory-motor regions (although perhaps that is best left to a follow-up manuscript).
- 4) What is the grounding of the discussion statement "While the left hemisphere sensory-motor cluster showed increased motor activity in response to the speech conditions..."? In 2.2.3 I saw a significant increase in motor activity in left versus right but not a comparison of left versus its own baseline.
- 5) Regarding the pattern of 8-30Hz findings for Auditory-only correct versus incorrect, I am wondering if an alternative interpretation is possible to your hypothesis that inhibition of the dorsal route may aid processing. It could be that the increases in sensory-motor areas observed during incorrect trials reflect a compensatory mechanism when words are not being processed well in the ventral stream. It may not be that the dorsal stream is interfering and in need of suppression but rather than it is not normally recruited unless there is difficulty in the ventral route, so activity is increased only on trials which were veering toward failure anyway.

Figures

1) Figure 1 is difficult to parse. For example, the ventral route always has an arrow that is sometimes crossed out. However, the black arrow for the dorsal route is missing in the first brain. Does the missing black arrow have a different meaning than the crossed out white arrow? Also, the underlying brain is low resolution and some of the interesting information (such as interconnections in each route) are obscured by the arrows. This is just a suggestion but I wonder if the figure could be better accomplished by one high-resolution model brain and with color-coded box-and-arrows for the presumed model activity for each of the four stimulus types (since the main information being communicated boils down to just two areas per stimulus type). Currently, the text does a good job of walking the reader through the model hypotheses but I found the figure too vague and cluttered.

2) Figure 2: Why use a hard trial for the example? The reader might be more easily able to grasp the superimposing of the stimulus waveform on the noise and the relationship between their onsets on an easy trial. Currently, the waveform appears to be all noise.

3) Figure 7: If possible, moving the bars closer together within stimulus type would help my eye more easily segregate the different stimulus types. Also, using the same color scheme to denote two different ways of dividing trials was a bit confusing to keep in mind.

4) Is it possible to mask time-frequency plots by significant difference from a baseline of 0? I found my eye being drawn to certain features but unable to tell if they were significantly different than chance. For instance, in Figure 6 on the left panel there is a patch of elevated power in the 8-12Hz range from ~100-500ms (which then transitions to a large decrease at ~800ms). Features such as this may be of interest to researchers but it is currently difficult to tell what is actually significant.

Minor:

1) Line 36: extra depending on in "...perceptual outcome depending on depending on the lexicality..."

2) Line 201: Reference to SI Table 3, but SI Table 3 is the number of independent components per condition. Is SI Table 4 meant?

3) What do you make of the cluster of 8-30Hz decrease in the right sensorimotor cluster related to response onset (Figure 5A right panel)? Since the participants were using their right hand, do you think this right-lateralized sensory-motor decreased activity is bleeding over from the left motor activity?

4) Why did the stimulus period used in analysis extend from -200ms prior to stimulus presentation instead of starting from a time >0ms?

5) It may be interesting information to include what the average volume was for each stimulus class to arrive at the 50%/80% score metrics (since I believe volume was the parameter varied to control difficulty?). Did one stimulus class prove to be reliably harder (i.e. needed a consistently higher volume across subjects) to achieve the same level of performance across participants?

6) Why was the right hand consistently used for responses instead of alternated to distribute motor activity equally across hemispheres or use the left hand since the right hemisphere sensorimotor areas would be a priori not expected to contribute much to speech perception?

Reviewer #2 (Remarks to the Author):

This paper aimed to investigate the role of the motor system in speech perception. Specifically, the authors wished to disentangle the effects of the stimulus and task conditions on engagement of the motor cortex during perception of whole words and meaningful non-speech sounds. They used an adaptive four-alternative forced choice task that isolates perception from decision-making and recorded EEG during this task. Specifically, they examined an EEG signature of motor activity (sensorimotor μ /beta suppression) during the perception of auditory phonemes, auditory words, audiovisual words, and environmental sounds, while holding difficulty constant at two levels (Easy/Hard). Authors believe their results revealed left-lateralized sensorimotor μ /beta suppression that was related to perception of speech but not environmental sounds. Audiovisual

word and phoneme stimuli showed enhanced left sensorimotor μ /beta suppression for correct relative to incorrect trials, while auditory word stimuli showed the opposite pattern: enhanced suppression for incorrect trials.

In terms of general comments, I am very familiar with the research topic area, and I think the authors' introduction and justification for the study and rationale is well-founded. They are indeed trying to disambiguate between numerous factors which could moderate engagement of the motor cortex during auditory/speech perception. Further understanding of these factors would make a nice contribution to the research area. However, I am not familiar with their EEG measure and I found their results difficult to interpret. Notably, this was not solely due to a lack of familiarity with the EEG measure, but more so how their statistics were presented. It was not clear to me that some of their statements were supported by the associated statistics. Results seemed somewhat unclear. Accordingly, it was difficult to judge the validity of the authors' claims in the discussion without a better understanding of the data. Whilst I am not an expert in the authors' EEG measure, I believe that a reader interested in this research area should be able to understand the data sufficiently to form an opinion on the conclusions posed. In this instance, I do not feel that the presentation of the results allows for this.

In addition, I have a number of specific comments for the authors:

1. What was the age-range of recruited participants?
2. Were participants monolingual?
3. What was the noise source in the excluded data? Exclusion of 7 datasets is almost 20% of the total recruited sample. More detail is required here.
4. Was an artefact criterion used online to reject noisy trials?
5. Why was pink noise chosen and how was it made? What were the characteristics?
6. How did the authors know that the 1000ms delay period was enough time to ensure that subvocal rehearsal, decision making, and button press response preparation were separated from stimulus identification?
7. Which packages were used in R? These need to be named and cited, and please add the model equation for the behavioural analysis.
8. Please add equations modelled in R for the EEG data.
9. Please add significance indicators to figures for clarity.
10. Please clarify the relevance of positive versus negative numbers in the EEG power measure, as individuals unfamiliar to this measure will not understand what 'enhanced suppression' should relate to numerically.
11. The authors state (line 198): 'The pattern of activity showed enhanced μ /beta suppression for correct relative to incorrect trials for the AVWords and Phonemes, while the AudWord stimuli showed the opposite pattern: enhanced suppression for incorrect trials (AudWord vs. AVWord, $t = 3.882$, $p < .001$; 200 AudWord vs. Phoneme, $t = 3.313$, $p = .001$).' Unless I have misunderstood something, it isn't clear why the statistics are not being presented for the correct vs incorrect comparisons for each trial type. It isn't clear why AudWords is being compared to all other trial types, but alternative trial comparisons are not being presented, or whether the trial types in the brackets with the statistics are the correct or incorrect values. This is very confusing and seems to disconnect the statements from the numerical statistics.
12. Line 206 'This was the only significant interaction, but given the pattern of results, we ran a separate model to investigate the main effects and found a significant main effect of stimulus type. The pattern of μ /beta suppression showed greater motor activity for AVWord and Phoneme stimuli than for the AudWord stimuli (AudWord vs AVWord, $t = -3.009$, $p = .003$; AudWord vs Phoneme, $t = -2.694$, $p = .007$).' this needs further justification as it reads as if the authors looked at the trends in the data and analysed according to what was observed, which is highly inadvisable.
13. What is the relevance of not crossing the zero axis? This needs to be clarified for readers who are unfamiliar with the technique.

Thank you for your thoughtful comments and suggestions on our manuscript, “Motor engagement relates to accurate perception of phonemes and audiovisual words, but not of auditory words.” We are grateful for both the positive comments as well as the constructive feedback. We have addressed each of your comments below, and the corresponding changes to the manuscript are highlighted in blue text. We appreciate all your time and effort and we feel your helpful insights have increased the clarity and quality of the manuscript.

Reviewer #1

Comment 1: The last analysis in 2.2.4 regarding the significance of power differences from baseline raises an interesting question. It appears that no condition across stimulus types would differ significantly from baseline after correction for multiple comparisons based on the p-values reported (all $p > .01$ with a high number of one-sample tests run). Does this surprise the authors? Does this suggest to the authors that the motor activity engaged during language perception is at a much lower power than would be engaged by overt action? It seems to make sense that the engagement of speech templates at a sub-movement level for speech perception would evoke much less activity than speech production even if generally the same neural populations are involved. You mention a previous study, [53], which focused on speech production using a similar paradigm. A comparison of overall magnitudes between production and perception would be of interest.

- The reviewer makes a very good observation. We did expect that the magnitude of activity evoked by listening to speech would be much lower than that evoked by speaking or preparing to produce overt or even covert speech. Because EEG is a relative measure and depends heavily on the baseline and time-frequency analysis used, it is difficult to compare EEG magnitudes across studies. However, it was surprising that in the study by Jenson and colleagues¹, their measures of phoneme perception and phoneme and word production appear to evoke similar magnitudes of μ /beta suppression. One possible contributor to this is that in this study, the same stimuli were repeated over and over, whereas in our study we used unique stimuli on almost every trial. Since the stimuli were very predictable in the Jenson study, it is possible that the subjects had pre-formed perceptuo-motor models, which could possibly contribute to the perception conditions evoking a magnitude of activity more similar to the production conditions. A discussion of magnitude of μ /beta suppression has been added to the Discussion (section 3.2, line 308).

Comment 2: It may be worth highlighting the lack of difference in 8-30Hz power between audiovisual words and phonemes in the broader discussion. Presumably the neural encoding of visual speech stimuli and the encoding of phonemic auditory speech stimuli are two different processes at circuit level. However, you demonstrate that in sensory-motor areas an overall similar amount of motor-related activity is evoked during processing of both stimuli. Do the authors believe the cognitive operations being performed in the audiovisual and phoneme trials are similar or do these trials just evoke similar amounts of overall neural activity?

- The reviewer brings up an interesting point. Our results show that these two stimulus types evoke comparable levels of motor activity, and according to motor modeling theories, it would be reasonable to think that both stimulus types engage similar motor modeling processes. However, this is an unanswered question, and future studies, such as those using multivariate classification techniques, may be better able to disentangle

any differences in the types of neural computations being performed during the perception of these two stimulus types. We have added this to the discussion (section 3.3.3, line 426).

Comment 3: Building off of #2, is it possible to get a temporal marker of the onset of significant 8-30Hz decreases from baseline for each stimulus type post stimulus presentation? Any temporal information would add additional constraints to interpretations as to how processing these different stimulus types proceeds in the sensory-motor regions (although perhaps that is best left to a follow-up manuscript).

- We agree that temporal information would be of interest. We designed our study to have unique stimuli on almost every trial in order to prevent participants from forming a perceptuo-motor model of an upcoming stimulus prior to presentation. While we believe this was an advantage, it may have also contributed to a high degree of variability in the temporal response profiles of our EEG measure, even within stimulus types. This variability is then combined with the inter-individual variability in peak μ /beta frequencies. Given this variability and the fact that our primary hypotheses were concerned with overall levels of engagement of the motor system to each stimulus type and difficulty level, for this study we chose to analyze the mean EEG responses within the stimulus presentation window. In the future, we plan to complete item-level analyses and look more specifically at temporal response profiles. We have noted this in the discussion (section 3.3.3, line 431).

Comment 4: What is the grounding of the discussion statement “While the left hemisphere sensory-motor cluster showed increased motor activity in response to the speech conditions...”? In 2.2.3 I saw a significant increase in motor activity in left versus right but not a comparison of left versus its own baseline.

- We thank the reviewer for this observation, and we have added an analysis (section 2.2.3, line 204) comparing the mean EEG activity across the analysis window in the left sensorimotor cluster against baseline, as well as masking both the left and right cluster time-frequency plots in Figure 5 by significant differences from baseline.

Comment 5: Regarding the pattern of 8-30Hz findings for Auditory-only correct versus incorrect, I am wondering if an alternative interpretation is possible to your hypothesis that inhibition of the dorsal route may aid processing. It could be that the increases in sensory-motor areas observed during incorrect trials reflect a compensatory mechanism when words are not being processed well in the ventral stream. It may not be that the dorsal stream is interfering and in need of suppression but rather than it is not normally recruited unless there is difficulty in the ventral route, so activity is increased only on trials which were veering toward failure anyway.

- We thank the reviewer for this insightful comment, and we have added this as an alternative interpretation in the discussion (section 3.3.3, line 401). However, by either interpretation, motor activity does not contribute to successful perception of auditory words, whereas it does appear to contribute to successful perception of audiovisual words and auditory-only phonemes.

Comment 6: Figure 1 is difficult to parse. For example, the ventral route always has an arrow that is sometimes crossed out. However, the black arrow for the dorsal route is missing in the first brain. Does the missing black arrow have a different meaning than the crossed out white arrow? Also, the underlying brain is low resolution and some of the interesting information (such as interconnections in each route) are obscured by the arrows. This is just a suggestion but I wonder if the figure could be better accomplished by one high-resolution model brain and with color-coded box-and-arrows for the presumed model activity for each of the four stimulus types

(since the main information being communicated boils down to just two areas per stimulus type). Currently, the text does a good job of walking the reader through the model hypotheses but I found the figure too vague and cluttered.

- Thank you for this feedback. We have remade Fig 1 (pg 5) to incorporate these suggestions.

Comment 7: Figure 2: Why use a hard trial for the example? The reader might be more easily able to grasp the superimposing of the stimulus waveform on the noise and the relationship between their onsets on an easy trial. Currently, the waveform appears to be all noise.

- In the original figure we did not actually have a stimulus waveform embedded within the noise, but we have added one to incorporate this feedback (Figure 2, pg 7).

Comment 8: Figure 7: If possible, moving the bars closer together within stimulus type would help my eye more easily segregate the different stimulus types. Also, using the same color scheme to denote two different ways of dividing trials was a bit confusing to keep in mind.

- We have edited Figure 7 (pg 15) accordingly.

Comment 9: Is it possible to mask time-frequency plots by significant difference from a baseline of 0? I found my eye being drawn to certain features but unable to tell if they were significantly different than chance. For instance, in Figure 6 on the left panel there is a patch of elevated power in the 8-12Hz range from ~100-500ms (which then transitions to a large decrease at ~800ms). Features such as this may be of interest to researchers but it is currently difficult to tell what is actually significant.

- We have added significance masks to both Figure 5 (pg 11) and Figure 6 (pg 12).

Minor Comments:

- 1) Line 36: extra depending on in "...perceptual outcome depending on depending on the lexicality..."
 - Fixed.
- 2) Line 201: Reference to SI Table 3, but SI Table 3 is the number of independent components per condition. Is SI Table 4 meant?
 - Yes, this has been fixed.
- 3) What do you make of the cluster of 8-30Hz decrease in the right sensorimotor cluster related to response onset (Figure 5A right panel)? Since the participants were using their right hand, do you think this right-lateralized sensory-motor decreased activity is bleeding over from the left motor activity?
 - We believe that rather than representing a bleeding of activity from the left hemisphere, the right sensorimotor cluster activity in Figure 5 reflects activity generated in the right hemisphere. This is in line with previous work showing that movement-related ERD occurs bilaterally, but is stronger on the contralateral side². We have noted this in the discussion (section 3.3.2, line 355).
- 4) Why did the stimulus period used in analysis extend from -200ms prior to stimulus presentation instead of starting from a time >0ms?
 - Previous work has shown that prestimulus EEG activity is related to perceptual performance, with alpha (or μ) power suppression indicating a release from inhibition and preparation for an upcoming stimulus^{3,4}, and prestimulus μ /beta suppression has been shown for correct trials during phoneme perception, perhaps signaling a ramping up of motor processing⁵. We included this

prestimulus period in our analysis window in order to incorporate preparatory activity. We have added this clarification to section 6.6.3, line 619.

- 5) It may be interesting information to include what the average volume was for each stimulus class to arrive at the 50%/80% score metrics (since I believe volume was the parameter varied to control difficulty?). Did one stimulus class prove to be reliably harder (i.e. needed a consistently higher volume across subjects) to achieve the same level of performance across participants?
 - We thank the reviewer for this helpful suggestion, and we have added an analysis of the volume (Results section 2.1, line 162 and Methods section 6.5, line 557) needed to maintain the appropriate difficulty levels.

- 6) Why was the right hand consistently used for responses instead of alternated to distribute motor activity equally across hemispheres or use the left hand since the right hemisphere sensorimotor areas would be a priori not expected to contribute much to speech perception?
 - The reviewer makes a good point that alternating hand would have been a nice approach, but instead we chose to control for button press-related motor activity by separating the response from stimulus presentation by a minimum of 1000ms. This allows us to look at both perception-related and overt motor-response related activity in the left hemisphere. This is discussed in section 6.3, line 520.

Reviewer #2

General comment: ...Further understanding of these factors would make a nice contribution to the research area. However, I am not familiar with their EEG measure and I found their results difficult to interpret. Notably, this was not solely due to a lack of familiarity with the EEG measure, but more so how their statistics were presented. It was not clear to me that some of their statements were supported by the associated statistics. Results seemed somewhat unclear. Accordingly, it was difficult to judge the validity of the authors' claims in the discussion without a better understanding of the data.

- We thank the reviewer for this helpful feedback, as well as the more detailed comments below. We have consulted with a statistician (author G. Norato), who ensured that the statistics were run and reported according to best practices. We have rewritten our methods and results sections to include more detail and hopefully provide a more straightforward presentation of our statistical analysis and results.

Additional comments:

1. What was the age-range of recruited participants?
 - The age range was 21 to 31 years. This has been added to section 6.1, line 485.

2. Were participants monolingual?
 - No, not necessarily. We felt it was most important to ensure that participants were exposed to English during development. Thus, we conducted language history screenings to ensure that each participant was a native English speaker, which we defined as learning English before the age of 5, having English as the primary language spoken growing up, and self-rating as fully proficient in reading, speaking, and understanding English. This means that participants could have also spoken another language besides English.

3. What was the noise source in the excluded data? Exclusion of 7 datasets is almost 20% of the total recruited sample. More detail is required here.

- Of these 7, two were excluded for having fewer than 8 independent components (see SI section 4, line 110), and the remaining 5 were excluded for having overall noisy or artefactual data. There was not one consistent source of noise across these subjects, but rather overall high numbers of artefacts identified in our combined manual and automatic data cleaning and artefact rejection steps. It is unclear why we saw this level of disruption in some subjects while others appeared relatively clean, but EEG data quality is known to be highly variable between subjects.

3. Was an artefact criterion used online to reject noisy trials?

- No. We used offline correction methods only (detailed in SI section 4, line 90).

5. Why was pink noise chosen and how was it made? What were the characteristics?

- Pink noise, also known as 1/f noise, refers to noise in which the power spectral density is inversely proportional to the frequency. Pink noise is commonly used as a mask in perceptual experiments^{6,7} because it is similar to the temporal envelope of speech⁸. The noise was generated in Audacity using the built-in noise generator before being sampled and RMS normalized to match the stimulus files. This information has been added to SI section 2, line 57.

6. How did the authors know that the 1000ms delay period was enough time to ensure that subvocal rehearsal, decision making, and button press response preparation were separated from stimulus identification?

- This delay period separated the perception of the stimulus from the presentation of the answer choices, and our EEG analysis parameters were confined to the stimulus presentation window, which ended prior to the 1000ms delay period. Response choices were not displayed until after the waiting period, meaning that the decision as to which of the 4 choices was correct could not begin until over 1000ms after the end of the stimulus presentation analysis window. In contrast to prior studies of speech perception in which the same small set of stimuli and answer choices occur over and over, our study also used unique stimuli and answer sets. This means that participants could not predict the stimulus or the answer choices, and thus could not form a perceptuo-motor model of the response options before answer choice presentation. This also means that the decision between options could not occur until answer choice presentation. As for the button press, it has been shown that button press response preparation typically ramps up just prior to a “go” signal or ready cue^{9,10}, and even in the case of extremely predictable movements requiring no decision or discrimination¹¹ (i.e. tapping to a beat), μ /beta suppression begins no more than one second prior to “go” or signal onset. As our analysis in 2.2.2 shows, our stimulus presentation window was separate from response-related activity. In our study the timing of subvocal rehearsal the timing is less clear, and the reviewer makes a valid point that we cannot rule out that subvocal rehearsal may have occurred. Studies looking at the timing of brain responses to phonemes have found phoneme category-specific responses in left premotor cortex/IFG areas as early as 120ms¹² and 144ms¹³ post-stimulus onset. However, these studies involved listening to repeated presentations of the same clearly presented phonemes or words, and this rapid time course reflects perception only, not preparation for production (covert or overt). Additional studies of electrophysiological responses during picture naming suggest that semantic/syntactical access when listening to words occurs closer to 400/500ms post-stimulus¹⁴. While we are unable to completely rule out that subvocal rehearsal could

have begun within our stimulus onset window, constraining our analysis to 1000ms post-stimulus onset does limit the contribution of repeated rehearsal to the observed activity. We thank the reviewer for pointing this out and have adjusted our claims accordingly (section 6.6.3, line 617).

7. Which packages were used in R? These need to be named and cited, and please add the model equation for the behavioural analysis.

- This information has been added to section 6.5 (multiple lines)

8. Please add equations modelled in R for the EEG data.

- This information has been added to section 6.6.3 (multiple lines)

9. Please add significance indicators to figures for clarity.

- We have added significance markers to all figures.

10. Please clarify the relevance of positive versus negative numbers in the EEG power measure, as individuals unfamiliar to this measure will not understand what 'enhanced suppression' should relate to numerically.

- Negative numbers in EEG power reflect a decrease, or suppression in power, whereas positive numbers reflect an increase in EEG power. EEG power over sensorimotor areas has been shown to be inversely related to cortical activity such that decreased power/enhanced suppression is related to greater motor activity. We discuss this relationship in section 1 and again in 3.2, line 281, and have added clarification throughout the text.

11. The authors state (line 198): 'The pattern of activity showed enhanced μ /beta suppression for correct relative to incorrect trials for the AVWords and Phonemes, while the AudWord stimuli showed the opposite pattern: enhanced suppression for incorrect trials (AudWord vs. AVWord, $t = 3.882$, $p < .001$; 200 AudWord vs. Phoneme, $t = 3.313$, $p = .001$).' Unless I have misunderstood something, it isn't clear why the statistics are not being presented for the correct vs incorrect comparisons for each trial type. It isn't clear why AudWords is being compared to all other trial types, but alternative trial comparisons are not being presented, or whether the trial types in the brackets with the statistics are the correct or incorrect values. This is very confusing and seems to disconnect the statements from the numerical statistics.

- We thank the reviewer for this feedback and we have rewritten the results section in an effort to clarify our statistical measures and findings. The mixed effects model we ran tests for a difference in difference (i.e. difference in correct vs. incorrect that is different across the condition types). In the revised text (line 216), before presenting the significant interaction effect, we first present the within-condition differences between correct and incorrect trials (these are the source of the interaction effect).

12. Line 206 'This was the only significant interaction, but given the pattern of results, we ran a separate model to investigate the main effects and found a significant main effect of stimulus type. The pattern of μ /beta suppression showed greater motor activity for AVWord and Phoneme stimuli than for the AudWord stimuli (AudWord vs AVWord, $t = -3.009$, $p = .003$; AudWord vs Phoneme, $t = -2.694$, $p = .007$).' this needs further justification as it reads as if the authors looked at the trends in the data and analysed according to what was observed, which is highly inadvisable.

- In our first model assessing accuracy across conditions we find a significant interaction in condition and accuracy, the pattern of which demonstrates that the activity evoked during the correct/incorrect AVWords and Phonemes displays an opposite pattern from that evoked during the AudWords. While we did not see a significant interaction between condition and difficulty, we wanted to explore whether the different pattern of activity between conditions (AVWords & Phonemes vs. AudWords) was also present. Thus, we conducted a model to specifically investigate any main effects of condition. We agree that because the overarching interaction was not significant, any main effects need to be considered descriptive, and we have added this caveat to the text (section 2.2.4, line 232).

13. What is the relevance of not crossing the zero axis? This needs to be clarified for readers who are unfamiliar with the technique.

- In this instance, the fact that none of the error bars cross the zero axis suggests that the activity evoked by the environmental sounds is not significantly different from baseline. We have added this clarification to the text (section 2.2.4, line 243).

REVIEWERS' COMMENTS:

Reviewer #1 (Remarks to the Author):

The authors have thoughtfully responded to all my comments, and I find the additions and modifications to the manuscript to address my concerns. I have no remaining major comments.

Reviewer #2 (Remarks to the Author):

The authors have addressed my comments and the statistics have been clarified in places. However, I still find it difficult to digest the dense results and how they cohere with the discussion. I suspect this difficulty is due to a combination of my lack of familiarity with the EEG measure used, and the complexity of the study design. Readers in a similar position (very familiar with the research area, less familiar with the EEG measures) may be happy to engage with the paper's findings as they appear in the discussion. However, the degree to which the reader's thinking may be influenced by the paper could be limited somewhat by how digestible the reader finds the paper's results.

As initially mentioned when I first reviewed this paper, the research topic is interesting and the rationale for the research is sound. I think the paper will be interesting to researchers working in related areas.

The authors' believe their main findings demonstrate that motor involvement in perception is left-lateralized; is specific to speech stimuli; and is not simply the result of domain-general processes. They believe this provides evidence for an interactive network for speech perception in which dorsal stream motor areas are dynamically engaged during the perception of speech depending on the characteristics of the speech signal.